# Joint Optimization of Pre-Marshalling and Yard Cranes Deployment in the Export Block

**Shuang Duan, Hongxing Zheng * and Xiaomin Gan**

School of Transport Engineering, Dalian Maritime University, Dalian 116026, China
* Correspondence: zhredstar@dlmu.edu.cn

**Abstract:** To improve the efficiency of loading operation by researching the optimization of the pre-marshalling operation scheme in the export container block between the time when the ship stowage chart was published and the beginning time of loading, a two-stage mixed integer programming model was established. The first stage established an optimization model of the container reshuffling location, based on the objective function of the least time-consuming operation of a single-bay-yard crane, and designed an improved artificial bee colony algorithm to solve it. Based on the first stage, an optimization model of yard crane configuration and scheduling was built to minimize the maximum completion time of the yard crane in the export block, and an improved genetic algorithm was designed to solve the built model. Through comparative analysis, the performance of our algorithm was better than CPLEX and traditional heuristic algorithms. It could still solve the 30 bays quickly, and the solving quality was 8.53% and 11.95% higher than GA and TS on average, which verified the effectiveness of the model and the science of the algorithm and could provide a reference for improving the efficiency of port operation.

**Keywords:** pre-marshalling; yard crane configuration; yard crane scheduling; mixed-integer programming

## 1. Introduction

In recent years, with the rapid growth of container port trade, the efficiency of loading operations at terminals has become a growing concern for all parties. A large number of relocations in the export container block seriously restricts the waiting time of the trucks in the block, which in turn affects the waiting time of the quay crane. It ultimately reduces the efficiency of the terminal's loading operations. For this reason, a reasonable and efficient pre-marshalling operation, as well as yard crane configuration and a scheduling plan, is an effective means to improve the efficiency of terminal loading operations, and an in-depth study is very necessary.

To improve the efficiency of loading operations, containers are stacked according to the ship's stowage chart as far as possible through pre-marshalling operations, thus reducing the number of relocations during loading. The focus of the research on pre-marshalling operations is on the sequence of retrieving and relocation. Tanaka et al. [1] designed an improved exact algorithm to solve for the best pre-marshalling operation solution. Based on this, a mathematical model was developed to eliminate the minimum number of reshuffles, a new branch-and-bound algorithm was proposed to solve it, and two new lower bounds were introduced (Tanaka et al. [2]). Ji et al. [3] introduced the ship's allocation plan in their study of the relocation problem. With the container-loading order determined, a mathematical model was established to minimize the number of relocations. Liu et al. [4] studied the quay crane double-loop problem considering container internal relocations and designed a polynomial-time heuristic algorithm. Boge et al. [5] studied pre-marshalling under uncertainty in container retrievals priority and introduced a new uncertainty model. Feng et al. [6] proposed a probabilistic model to describe the randomness of truck arrival

and built and solved a stochastic dynamic model to minimize relocations. da Silva et al. [7] studied the pre-marshalling problem with a determined order of pickups, proposing a new integer programming model to minimize the relocation cost.

For the optimization problem of pre-marshalling considering yard crane operation, Parreno-Torres scholars [8–10] conducted a great deal of research. Firstly, they studied the problem of pre-marshalling using the idle time before ship arrival and designed and tested an integer linear model to solve it. On this basis, they proved that the minimum number of relocations after pre-marshalling does not represent the shortest operating time of the yard crane. Therefore, a pre-marshalling operation optimization problem to minimize the operation time of the yard crane during loading was studied.

It can be seen that the configuration and scheduling of the yard crane had a significant impact on the efficiency of the yard's relocation operations. Li et al. [11] used a rolling time-domain algorithm to study the multiple-yard crane scheduling problem within a single block. On the other hand, Wu et al. [12], based on the work of Li et al. [11], considered the number of delayed tasks in retrieval in the objective function. Chu et al. [13] studied the scheduling and allocation of multiple yard cranes in two adjacent container blocks. Speer et al. [14] discussed the constraints in detail, such as yard crane movement interference, through a simulation model and proposed the branch-and-bound method to solve the real-time scheduling problem of the yard crane. Jin et al. [15] investigated the optimization problem of space allocation and yard crane deployment decisions and constructed an integer linear programming model to specify the working area of yard cranes in each period. Zweers et al. [16,17] used the idle time of the yard crane for pre-marshalling operations to reduce future container movements and divided it into two phases, namely, retrieval and relocation.

It is well known that the pre-marshalling problem is an NP-hard problem that can be solved by both exact (Tanaka et al. [1]) and heuristic algorithms. However, the general accurate algorithms cannot solve large-scale experiments. For more applications of heuristic algorithms, see Lee and Hsu [18], Lee and Chao [19], Bortfeldt and Forster [20], Huang and Lin [21], Tus et al. [22], Gharehgozli et al. [23], Jovanovic et al. [24], and Gheith et al. [25]. In recent years, various algorithms have been developed and applied to solve pre-marshalling and yard crane scheduling problems, including genetic algorithms (Hottung and Tierney, [26]), target-guided algorithms (Wang et al. [27]), target-driven algorithms (Wang et al. [28]), A* and IDA* algorithms (Ha [29], Tierney et al. [30]), etc. Lersteau et al. [31] reviewed the optimization methods for the pre-marshalling problem and distinguished four categories of optimization methods. Hottung et al. [32] even combined deep learning with heuristic algorithms, namely a Deep Learning Heuristic Tree Search (DLTS), and demonstrated that DLTS provided the highest-quality heuristic solution to date for the container pre-marshalling problem. The most commonly used exact algorithm is the branch and bound algorithm. Jin and Yu [33] discovered an over-pruning issue that occurs in the iterative deepening branch-and-bound algorithm by Tanaka and Tierney [1], and they proposed the lexicographic dominance principle to solve the problem. On this basis, Jin and Tanaka [34] collaborated on a paper on the unrestricted container relocation problem and designed an efficient iterative deepening branch-and-bound algorithm.

In summary, there is a growing body of literature on pre-marshalling operations, and certain results have been achieved, but there are still some limitations. Most of the existing literature was in pursuit of the minimum number of relocations during loading, only a small amount of literature has considered the scheduling and operation time of the yard crane but did not involve the configuration of the number of yard cranes, and rarely do researchers integrate the relocation sequence and relocation position optimization. Therefore, this paper focuses on the optimization of pre-marshalling operation and yard crane deployment in the container terminal during the pre-marshalling period. By reasonably arranging the appropriate number of yard cranes and formulating the optimal scheduling plan, the pre-marshalling operation is carried out in this block with the shortest yard cranes' operation time to improve the efficiency of the loading in the block.

The main contributions of this study are as follows:

(1) By combining the yard cranes' configuration and scheduling, the joint optimization of the pre-marshalling operation and the yard cranes deployment was realized, and a two-stage mixed integer programming model was constructed.
(2) We designed an improved artificial bee colony (IABC) algorithm to solve the problem based on the reshuffling model.
(3) We designed an improved genetic algorithm (IGA) to solve the problem based on the yard crane configuration and scheduling model.
(4) Targeted at the above problems and models, the proposed algorithm was compared with other heuristic algorithms to verify the effectiveness and superiority of the proposed algorithm.

The remainder of this paper is organized as follows. Section 2 presents the problem description and mathematical modeling. Section 3 introduces the design of the IABC algorithm and the IGA. The experiments and discussion are conducted in Section 4. Section 5 presents the conclusions of the study and suggestions for future research.

## 2. Problem Description and Mathematical Modeling

### 2.1. Problem Description

Considering that most of the container terminals in China currently have limited yard space, the unused space available for pre-marshalling is very scarce, so pre-marshalling is abandoned by most of the yards because it is too complicated. Therefore, pre-marshalling within the initial bay of the original export container block is the current common pre-marshalling mode.

If the containers of the designated ship are called target containers, they are not arranged in the order of loading and need to be relocated. In Figure 1, we provide an example solution where the relocation step (a, b) means the top container of stack a is relocated to the top empty container space allocation of stack b, just as the relocation step (5, 6) means the top container of stack 5 is relocated to the top empty container space allocation of stack 6.

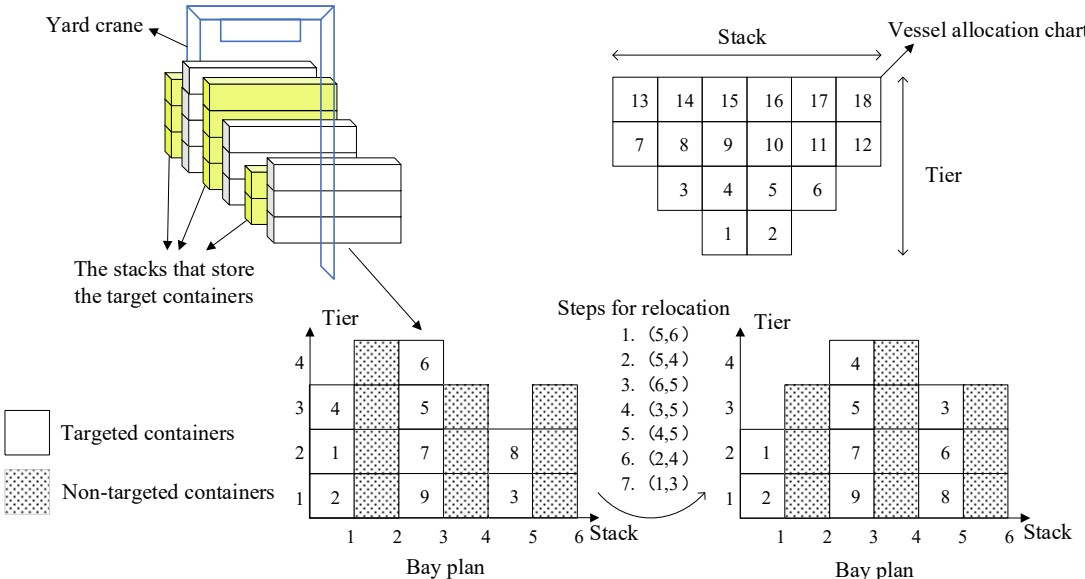

**Figure 1.** Pre-marshalling process.

A container block consists of multiple bays, each of which contains multiple stacks. After investigations at Tianjin Port and other places, it was found that containers of different ships are usually stored in adjacent stacks for better loading operation. Therefore, target containers cannot be placed in the container stacks storing other ships after the

pre-marshalling is completed. As shown in Figure 1, target containers are stored in odd stacks, and non-target containers are stored in even stacks. Each stack can hold more than one container, but there is a height limit. In addition, it is not allowed to relocate over the height to avoid security risks. As shown in Figure 2, Container No. 2 on the first stack cannot be relocated to the top of the third stack, that is, steps (1, 3) cannot be performed.

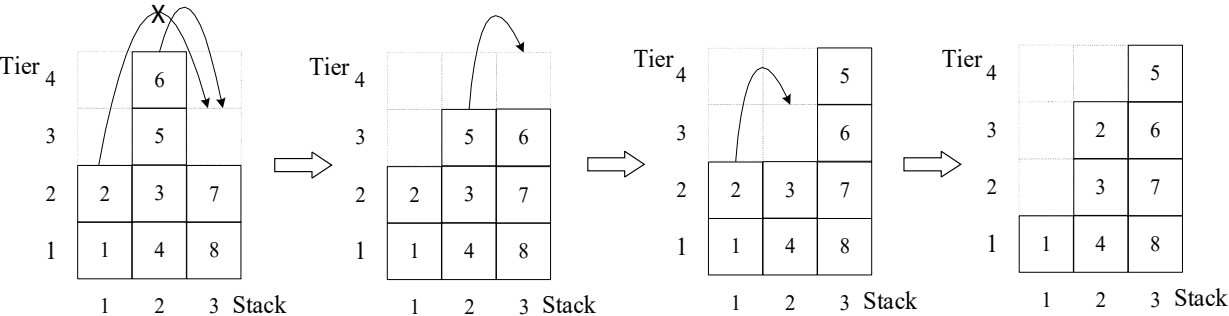

**Figure 2.** Relocation process.

The number of yard cranes configured in each container block can be selected according to the length of the pre-marshalling period. During operation, practical restrictions such as the safe distance between yard cranes and non-crossing of yard cranes should be taken into account.

### 2.2. Mathematical Modeling

In the actual process of container terminals, for safety reasons, container relocation in the yard is generally carried out only within a single bay, rarely across bays, and the relocation operations within each bay are not affected by other bays. Therefore, when studying the optimization of the yard crane configuration and scheduling in the pre-marshalling operation, each bay can be considered as a whole, and it is not necessary to comprehensively consider the sequence of relocation of all containers in the block. It can be seen that the relocation in the bay has little influence on the order of the bay for the yard crane operation. Therefore, the problem in this paper is divided into two parts. Firstly, the first stage determines the reshuffle in each bay. Then, based on this, the second stage determines the order of yard cranes' operations. This can simplify the problem, as modeling and solving are clearer and more convenient.

#### 2.2.1. Model Assumption

The reshuffling optimization model and the yard crane configuration and scheduling model proposed by us are common in the literature. There are certain common assumptions, for example, all containers are of the same type (e.g., 20 or 40 foot) (Tanaka et al. [1]) and the crane can access the bay only from the top and containers are either relocated to the top of another stack or the ground (Azab et al. [35]). However, we discuss certain implicit assumptions here. A new term—the pre-marshalling period—is proposed in this paper, which refers to the period from the time when the target ship's allocation chart is published to the time when the container block starts loading. This period is known in advance. In addition, the second assumption is that the target containers are all stored in odd-numbered stacks in the export container block to be shipped. According to the field investigation of Tianjin Port and others, containers of different ships are stored in adjacent stacks in a bay, which can save a lot of time in the loading operation.

#### 2.2.2. Mathematical Notations

(1) Parameters

$L$: Collection of all bays where containers are to be loaded.
$S_l$: Collection of all containers in $l^{th}$ bay.

$N_l$: Collection of target containers in $l^{th}$ bay.

$N_l^-$: Collection of non-target containers in $l^{th}$ bay.

$W$: Collection of stacks in a bay.

$H$: Collection of tiers in a stack.

$K$: Collection of all yard cranes.

$L_0$: Total number of bays where containers are to be loaded.

$H_0$: Maximum number of containers that can be stored in a stack.

$K_0$: Total number of yard cranes.

$t_0$: Average relocation time for yard crane.

$D$: Safe distance between yard cranes (bay).

$t_1$: Walking a bay for the yard crane.

$M$: A fairly large number.

(2) Decision variables

Phase 1 decision variables

$a_{nwh}$: If container n is at storage $(w, h)$, it is 1; otherwise, it is 0.

$b_{nw}$: If the stack $w$ on which container $n$ is located in an even stack, it is 1; otherwise, it is 0.

$c_{wij}$: If container $i$ is below container $j$ in the stack $w$, it is 1; otherwise, it is 0.

$x_{nww_1}$: If the yard crane relocates container $n$ from stack $w$ to stack $w_1$, it is 1; otherwise, it is 0.

$y_{wij}$: If the retrieval order of container $i$ is before container $j$ in stack $w$, it is 1; otherwise, it is 0.

$g_{ij}$: If the yard crane operates container $i$ first and container $j$ immediately after, it is 1; otherwise, it is 0.

$T_l$: Total operating time of the yard crane at $l^{th}$ bay.

Phase 2 decision variables

$d_{lk}$: If the yard crane k operates bay $l$, it is 1; otherwise, it is 0.

$f_{\alpha\beta k}$: If the yard crane $k$ operates bay $\alpha$ first and bay $\beta$ immediately after, it is 1; otherwise, it is 0.

$e_{lk}$: If bay $l$ is the first operation of yard crane $k$, it is 1; otherwise, it is 0.

$m_{lk}$: If bay $l$ is the last operation of yard crane $k$, it is 1; otherwise, it is 0.

$T_k$: Total operating time for yard crane $k$.

(3) State variables

$p_{ijkk'}$: If yard crane $k$ is operating at bay $i$ and also yard crane $k'$ is operating at bay $j$, it is 1; otherwise, it is 0.

$ST_n$: Start moment of container $n$ for yard crane operations.

$FT_n$: Completion moment of container $n$ for yard crane operations.

$BT_l$: Start moment of bay $l$ for yard crane operations.

$ET_l$: Completion moment of bay $l$ for yard crane operations.

### 2.2.3. Mathematical Formulation

We constructed a two-stage mixed integer programming model. The first stage is to determine the current stacking status and loading order of each target container in a certain export container block and study the optimal reshuffle of each in-bay target container so that the yard crane can complete the pre-marshalling at that bay in the shortest time. The second stage is based on the optimal pre-marshalling scheme of each bay obtained in the first stage. We study the yard crane configuration and scheduling optimization scheme in the block, and, finally, give the optimal number and order of the yard crane configuration.

(1) Optimization model for the reshuffle

According to the characteristics of the first stage of the problem, the optimization objective is to minimize the operation time of the yard crane relocation.

$$Min\,z = T_l \;\; \forall l \in L \tag{1}$$

Taking a single bay as the object of study and considering the limiting effects of yard crane relocations and the effects of storage of containers from different vessels can be expressed as

$$a_{nwh+1} \leq a_{nwh} \ \forall n \in S_l, w \in W, h \in \{1, \ldots, H_0 - 1\} \tag{2}$$

$$\sum_{n \in S_l} a_{nwh+1} \leq \sum_{n \in S_l} a_{nwh} \ \forall w \in W, h \in \{1, \ldots, H_0 - 1\} \tag{3}$$

$$\sum_{n \in S_l} a_{nwh} \leq 1 \ \forall w \in W, h \in H \tag{4}$$

$$\sum_{w \in W} \sum_{h \in H} a_{nwh} = 1 \ \forall n \in S_l \tag{5}$$

$$\sum_{n \in S_l} \sum_{h \in H} a_{nwh} \leq H_0 \ \forall w \in W \tag{6}$$

$$\sum_{w_1 \in W \backslash \{w\}} x_{nww_1} \leq 1 \ \forall n \in S_l, w \in W \tag{7}$$

$$\sum_{i \in S_l} \sum_{h \in H} a_{iwh} - \sum_{i \in S_l} c_{win} - 1 \leq (1 - x_{nww_1}) \times M \ \forall n \in S_l, w \in W, w_1 \in W \backslash \{w\} \tag{8}$$

$$\sum_{i \in S_l} \sum_{h \in H} a_{iw_1h} + x_{nww_1} \leq H_0 \ \forall n \in S_l, w \in W, w_1 \in W \backslash \{w\} \tag{9}$$

$$\sum_{i \in S_l} \sum_{h \in H} a_{iw_2h} \leq \sum_{i \in S_l} \sum_{h \in H} a_{iwh} + (1 - x_{nww_1}) \times M \ \forall n \in S_l, w \in W, w_1 \in W \backslash \{w\}, w_2 \in \{\min(w, w_1) + 1, \cdots, \max(w, w_1) - 1\} \tag{10}$$

$$\sum_{i \in S_l \backslash \{j\}} g_{ij} \leq 1 \ \forall j \in S_l \tag{11}$$

$$\sum_{i \in S_l \backslash \{j\}} g_{ji} \leq 1 \ \forall j \in S_l \tag{12}$$

$$b_{nw} \leq x_{nww_1} \times M \ \forall n \in N_l, w \in W, w_1 \in W \backslash \{w\} \tag{13}$$

$$1 - b_{nw} \leq x_{nww_1} \times M \ \forall n \in N_l^-, w \in W, w_1 \in W \backslash \{w\} \tag{14}$$

$$ST_n + t_0 \leq FT_n + (1 - x_{nww_1}) \times M \ \forall n \in S_l, w \in W, w_1 \in W \backslash \{w\} \tag{15}$$

$$FT_i \leq ST_j + (1 - g_{ij}) \times M \ \forall i \in S_l, j \in S_l \backslash \{i\} \tag{16}$$

$$\min \left( \sum_{w \in W} \sum_{i \in N_l} \sum_{j \in N_l \backslash \{i\}} c_{wij} \times y_{wij} \right) \tag{17}$$

$$T_l \geq FT_i - ST_j \ \forall i \in S_l, j \in S_l \backslash \{i\}, l \in L \tag{18}$$

Constraints (2) and (3) ensure that containers cannot be placed overhead. Constraint (4) ensures that only a maximum of one container can be stored in each storage. Constraint (5) ensures that a container can only occupy one storage. Constraint (6) ensures that the height of each stack in a bay must not exceed the maximum number of tiers. Constraint (7) ensures that a container can be relocated into, at most, one stack. Constraint (8) ensures that the container to be relocated can only be the top container in each stack. Constraint (9) ensures that a container can only be relocated to the upper-most tier of a stack and cannot exceed the maximum number of tiers. Constraint (10) ensures that a yard crane cannot cross height to relocate. Constraint (11) ensures that a container has, at most, one container working immediately in front of it. Constraint (12) ensures that a container has, at most, one container working immediately behind it. Constraint (13) ensures that the final target containers are to be placed in odd-numbered stacks. Constraint (14) ensures that all final non-target containers are to be placed in even-numbered stacks. Constraint (15) ensures the relationship between containers' start and finish times for yard crane operations. Constraint (16) ensures the relationship between the sequence

of containers and the yard crane's operation time. Constraint (17) ensures that the target container is guaranteed to have the least number of overlapped containers during the operation. Constraint (18) ensures the total operating time of the yard crane at bay *l*.

(2)    Optimization model for yard crane configuration and scheduling

According to the characteristics of the second stage of the problem, the optimization objective is to minimize the maximum completion time of the yard crane in the export block.

$$Min \, z' = \max\{T_k\} \, \forall k \in K \tag{19}$$

Taking a single block as the object of study and considering the effects of uncrossable and crossover operations between yard cranes, etc., can be expressed as

$$\sum_{k \in K} d_{lk} = 1 \, \forall l \in L \tag{20}$$

$$\sum_{l \in L} \sum_{k \in K} d_{lk} = L_0 \tag{21}$$

$$\sum_{l \in L} e_{lk} = 1 \, \forall k \in K \tag{22}$$

$$\sum_{l \in L} \sum_{k \in K} e_{lk} = K_0 \tag{23}$$

$$d_{lk} \geq e_{lk} \, \forall l \in L, k \in K \tag{24}$$

$$\sum_{l \in L} m_{lk} = 1 \, \forall k \in K \tag{25}$$

$$\sum_{l \in L} \sum_{k \in K} m_{lk} = K_0 \tag{26}$$

$$d_{lk} \geq m_{lk} \, \forall l \in L, k \in K \tag{27}$$

$$e_{lk} + m_{lk} \leq 1 \, \forall l \in L, k \in K \tag{28}$$

$$\sum_{\alpha \in L \setminus \{\beta\}} f_{\alpha\beta k} \leq 1 \, \forall \beta \in L, k \in K \tag{29}$$

$$\sum_{\alpha \in L \setminus \{\beta\}} f_{\beta\alpha k} \leq 1 \, \forall \beta \in L, k \in K \tag{30}$$

$$\sum_{\alpha \in L \setminus \{\beta\}} f_{\alpha\beta k} = d_{\beta k} - e_{\beta k} \, \forall \beta \in L, k \in K \tag{31}$$

$$\sum_{\alpha \in L \setminus \{\beta\}} f_{\beta\alpha k} = d_{\beta k} - m_{\beta k} \, \forall \beta \in L, k \in K \tag{32}$$

$$2 \times f_{\alpha\beta k} \leq d_{\alpha k} + d_{\beta k} \, \forall \alpha \in L, \beta \in L \setminus \{\alpha\}, k \in K \tag{33}$$

$$ET_l \geq BT_l + T_l \, \forall l \in L \tag{34}$$

$$BT_\beta \geq ET_\alpha + |\alpha - \beta| \times t_1 + M \times \left(f_{\alpha\beta k} - 1\right) \, \forall \alpha \in L, \beta \in L \setminus \{\alpha\}, k \in K \tag{35}$$

$$2 \times p_{\alpha\beta kk'} \leq d_{\alpha k} + d_{\beta k'} \, \forall \alpha \in L, \beta \in L \setminus \{\alpha\}, k \in K, k' \in K \setminus \{k\} \tag{36}$$

$$|\alpha - \beta| \geq D + M \times \left(p_{\alpha\beta kk'} - 1\right) \, \forall \alpha \in L, \beta \in L \setminus \{\alpha\}, k \in K, k' \in K \setminus \{k\} \tag{37}$$

$$\left(ET_\alpha - BT_\beta\right) \times \left(BT_\beta - BT_\alpha\right) \geq M \times \left(p_{\alpha\beta kk'} - 1\right) \, \forall \alpha \in L, \beta \in L \setminus \{\alpha\}, k \in K, k' \in K \setminus \{k\} \tag{38}$$

$$T_k \geq ET_\beta - BT_\alpha + (e_{\alpha k} - 1) \times M + \left(m_{\beta k} - 1\right) \times M \ \forall \alpha \in L, \beta \in L \backslash \{\alpha\}, k \in K \quad (39)$$

Constraint (20) ensures that a bay can only be operated by one yard crane. Constraint (21) ensures that the total number of bays operated by the yard cranes is the sum of the bays where the containers to be loaded are located. Constraints (22)–(24) ensure that each yard crane can only have one first operating bay. Constraints (25)–(27) ensure that each yard crane can only have one last operating bay. Constraint (28) ensures that the first and last operations cannot be the same for each yard crane. Constraint (29) ensures that a bay has, at most, one bay of the immediately preceding operation. Constraint (30) ensures that a bay has, at most, one bay for the immediately following operation. Constraint (31) ensures that the first operating bay of each yard crane has no bays immediately in front of it. Constraint (32) ensures that the last operating bay of each yard crane has no bays behind it. Constraint (33) ensures that the sequence of operations is guaranteed for the yard cranes. Constraint (34) ensures the relationship between the start and finish times of the yard crane operations at the lth bay. Constraint (35) ensures the sequence and time relationship of the yard crane operations. Constraint (36) ensures the operating relationship between the yard cranes. Constraints (37) and (38) ensure that a safe distance is maintained between the yard cranes and that they are not to be crossed. Constraint (39) ensures the total operating time for yard crane *k*.

## 3. Algorithm Design

The problem belongs to a two-stage optimization problem for yard crane pre-marshalling operations and configuration scheduling, for which a two-stage mathematical programming model is constructed. As each stage belongs to the NP-Hard problem, a two-stage heuristic algorithm is designed for solving it: The first-stage algorithm is used to solve the optimization model for the reshuffle to obtain the shortest operation time of the yard crane and the last storage location of the target containers; the second-stage algorithm is used to solve the optimization model for yard crane configuration and scheduling to obtain the optimal solution of the yard crane configuration and scheduling using the optimal solution obtained in the first stage as a known condition.

### 3.1. Design of IABC Algorithm

3.1.1. Pseudocode

The mathematical model of the first stage is a non-linear integer programming model, so this paper designs an improved artificial bee colony algorithm for solving it, which is better in terms of global convergence and merit-seeking ability. The artificial bee colony (ABC) algorithm is a popular metaheuristic that was originally conceived for tackling continuous function optimization tasks (Aydm et al. [36]). Table 1 shows the parameters in the IABC Algorithm pseudo-code (Algorithm 1).

3.1.2. Generation of Initial Populations

The first stage of this paper seeks to ensure that the minimum number of overlapped containers is achieved in the least amount of time by relocations. The location of a honey source indicates a feasible solution to the problem, the process of solving the optimal solution is the process of finding the best honey source, and the number of feasible solutions is equal to the number of honey sources.

**Algorithm 1** IABC Algorithm Pseudocode

1: Initialization: $iter \leftarrow 0$, $i \leftarrow 0$, $l \leftarrow 0$
2: Generate initial population containing pop feasible solutions according to Section 3.1.2
3: Evaluate each honey source in the population, recording the optimal honey source $x_{best}$ and $f$ ($x_{best}$)
4: **do until** $iter > Gen$
5:   Move the employed bees onto their honey sources according to Section 3.1.3
6:   Calculate the value of the objective function for the honey source in the population
7:   Onlookers select the new honey source $x_i$ based on probability $p_i$
8:   **if** $l < L$ **then**
9:     Neighborhood search by scout bees according to Section 3.1.4 for the new honey source $x'_i$
Set $l \leftarrow l + 1$
10:       if $f(x'_i) < f(x_i)$ **then**
11:             $x_i \leftarrow x'_i$
12:       **end if**
13:   else ($f(x'_i) > f(x_i)$) **do**
14:       Randomly generate a solution $x^*_i$ in the population to replace $x_i$
15:   **end if**
16:   Perform the genetic verification operation on populations according to Section 3.1.5
17:   Update the global optimal solution $x_{best}$ and $f(x_{best})$
18: **loop**
19: **return** $x_{best}$ and $f(x_{best})$

**Table 1.** Parameters of the IABC Algorithm.

| Symbol | Meaning |
| --- | --- |
| *Gen* | Total number of generations |
| *pop* | Initial population size |
| *L* | Limit on the number of updates to the honey source |
| *N* | Target containers collection |
| $p_i$ | onlookers selection probability |

The target containers are numbered from smallest to largest in accordance with the order of loading, and the code of the solution represents the position of the target container in the bay position after the end of reshuffling, i.e., the stack serial number and tier, whereby the first layer of the code is the stack where the target container is located and the second layer is the tier where the target container is located.

Since the quality of the initial solution greatly impacts the speed and quality of the solution of the IABC algorithm, this paper directly generates the solution with the smallest number of overlapped containers, and the specific strategy is as follows:

Step 1: Determining the population size, where *i* denotes the number of individuals in the population and *i* = 1.

Step 2: Determining the number of target containers *N*, the set *W* of odd stacks, and randomly generating *N* elements in the set *W*, denoted as set *A*.

Step 3: Determining the highest tier of the stack is *H* and whether there is an element in the set *A* whose number exceeds *H*; if it exists, go to Step 4; otherwise, set *A* is the first level of encoding; go to Step 5.

Step 4: Recording the number of elements exceeding *H*; the number of exceedances is given as *c*; we replace the position of these *c* elements in the set *A*, select the odd stack that stores the least number of target containers for replacement, and set *A* obtained is the first level of coding.

Step 5: Generating a set *B* of length *N*, recording the number *S* and location of different elements in set *A*, and generating the integers starting from *S* in order minus 1 at these corresponding positions in set *B*; set *B* is the second level of encoding.

Step 6: *i* = *i*+1, returning to Step 2 until *i* is greater than the population size, stopping iteration.

### 3.1.3. Employed Bees Search

Employed bees perform improved searches for honey sources. The second level of codes in the solution is derived from the first level of codes, so the cross-variation operation is only for the first level of codes, and the second level of codes is regenerated after the genetic verification operation.

The crossover operation uses the RPX crossover operator, as shown in Figure 3a, to swap the elements in m1 and m2 at the corresponding positions, leaving the rest unchanged, to obtain the new offspring m1′ and m2′; the variation operation uses the replacement variation, as shown in Figure 3b.

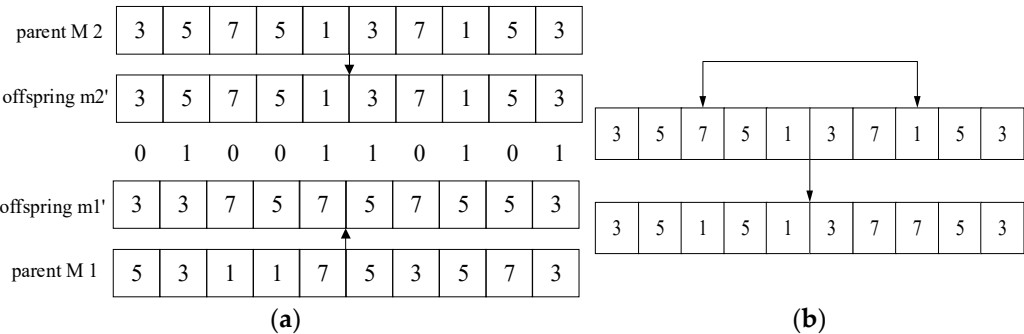

**Figure 3.** Employed bees search operations. (**a**) Schematic diagram of exchange; (**b**) schematic diagram of mutation.

### 3.1.4. Scout Bees Search

By designing a new local search operation, a new honey source is generated and compared with the objective function value of the honey source within the original population. If the value of the new source is smaller, the original source is replaced; if the value of the original source is smaller, the new source is discarded; if the two values are equal, the replacement is carried out with a fixed probability. The process is shown in Figure 4.

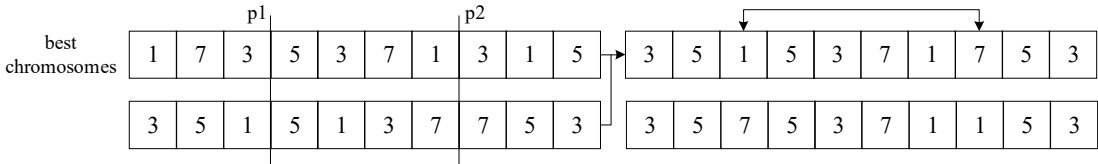

**Figure 4.** Schematic diagram of scout bees' search operation.

### 3.1.5. Genetic Verification Operation

Step 1: Determining whether the honey source is a feasible solution, the maximum number of tiers in the stack is recorded as $H$, judging whether the number of an element in the first level of the code exceeds $H$; if so, go to Step 2; otherwise, go to Step 3, generating the second level of the code.

Step 2: Determining the number of elements exceeding $H$, where the excess number is recorded as $c$; selecting the position of the element whose number is $c$ in the first-level code for replacement, giving priority to the odd stack with the least number of target containers to obtain the new first-level code.

Step 3: Generating a set whose length is the number of target containers, recording the number $S$ and position of different elements in the first level of encoding, and generating the integers starting from $S$ in order minus 1 at the corresponding element position in the set to obtain the set that is the second layer of encoding.

### 3.2. Design of IGA Algorithm

The second stage is the configuration and scheduling optimization problem of the yard cranes. The model is a non-linear integer programming model, as can be seen from the realistic constraints of not crossing between the yard cranes and maintaining safe distances, so an improved genetic algorithm (IGA) is designed to solve it, with the process shown in Figure 5.

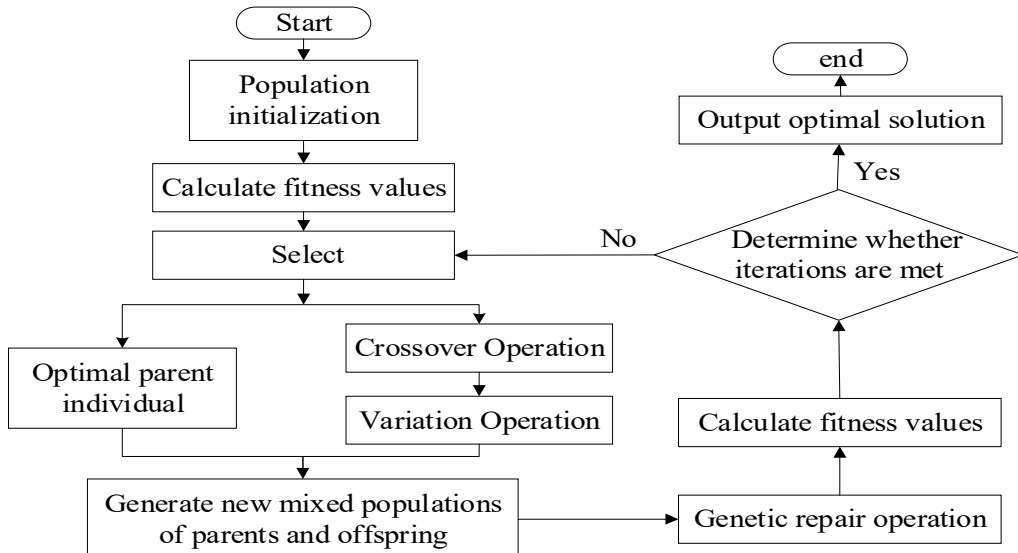

**Figure 5.** IGA flowchart.

#### 3.2.1. Encoding and Decoding

Based on the characteristics of the second-stage problem, two levels of coding are designed. The first level of coding from left to right is the order of the yard crane operations in terms of the bay and the second level of coding indicates the yard crane operations assignment. Figure 6 shows the coding schematic of a feasible solution, i.e., a solution for an export block operation with 15 bays and three yard cranes, where 0 indicates operation by yard crane No. 1, 2 indicates operation by yard crane No. 2, and 1 indicates operation by yard crane No. 3, where bays. 2, 3, 1, 4, and 5 are operated by yard crane No. 1 in sequence, bays. 8, 6, 10, 9, and 7 are operated by yard crane No. 2 in sequence, and bays 14, 13, 11, 12, and 15 are operated in sequence by yard crane No. 3.

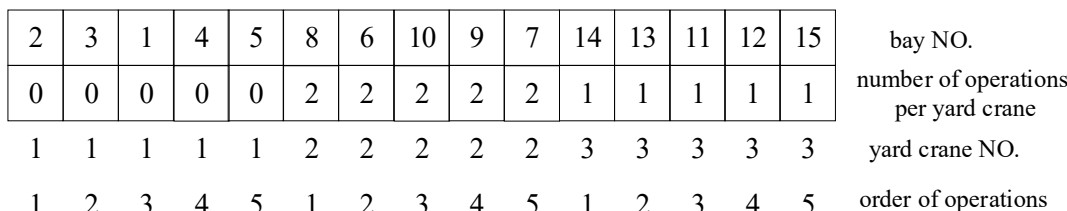

| 2 | 3 | 1 | 4 | 5 | 8 | 6 | 10 | 9 | 7 | 14 | 13 | 11 | 12 | 15 | bay NO. |
|---|---|---|---|---|---|---|----|---|---|----|----|----|----|----|---------|
| 0 | 0 | 0 | 0 | 0 | 2 | 2 | 2 | 2 | 2 | 1 | 1 | 1 | 1 | 1 | number of operations per yard crane |
| 1 | 1 | 1 | 1 | 1 | 2 | 2 | 2 | 2 | 2 | 3 | 3 | 3 | 3 | 3 | yard crane NO. |
| 1 | 2 | 3 | 4 | 5 | 1 | 2 | 3 | 4 | 5 | 1 | 2 | 3 | 4 | 5 | order of operations |

**Figure 6.** Chromosome coding.

#### 3.2.2. Population Initialization

To ensure the quality of the feasible solutions generated, firstly, the operating yard cranes are selected according to the serial number of bays. Since there is no crossover between the yard cranes, the preceding bays are assigned to the first yard crane, and the following bays are assigned to the last yard crane. Then to judge whether the solution within the population is feasible, i.e., when the yard cranes are crossed, the operating bays are switched; when the distance between the yard cranes does not meet the safety requirement in the case, the bays that satisfy the conditions are retrieved and replaced to generate a feasible initial population.

### 3.2.3. Crossover Operation

According to the problem and chromosomal characteristics, the sequential crossover method was used to generate new individuals via crossover recombination of chromosome codes, as shown in Figure 7. Two crossover points, p1 and p2, are randomly selected in the parent chromosomes M1 and M2, and the gene fragments between the parent chromosomes p1 and p2 are retained to obtain two incomplete offspring individuals, m1 and m2. The same gene codes of m2 and m1 are deleted from the parent chromosomes M1 and M2, respectively, and the remaining gene codes are sequentially arranged to obtain m1′ and m2′. The first p1 gene codes of m2′ and m1′ are sequentially inserted on the left side of m1 and m2, respectively. The remaining gene codes in m2′ and m1′ are inserted sequentially on the right side of m1 and m2, respectively, to form the new offspring chromosome. By retrieving the second level of coding of new individuals, bays with the same yard crane operations are placed together. In Figure 7, the red 6 and 8 switch positions, and the corresponding 2 and 1 repeat the operation. If there is a reduction in the number of yard cranes in the second layer code during the crossover, the chromosome is deleted and an alternative chromosome is randomly selected for duplication.

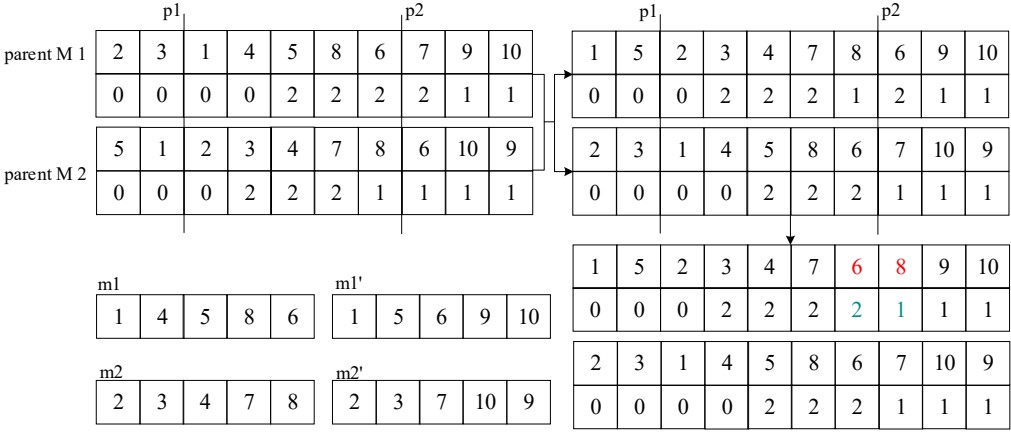

**Figure 7.** Chromosome crossover diagram.

### 3.2.4. Variation Operation

According to the characteristics of chromosome two-level coding, two steps are designed to perform the variation operation, i.e., the variation of inter-yard crane operation and the variation of the order of each yard crane operation, as shown in Figure 8. For a chromosome, firstly, a real number between 0 and 1 is randomly generated, and when this number is smaller than the variation probability *r1*, then a variation of inter-yard cranes operation is performed, i.e., one bay from each yard crane operation is randomly selected to be replaced with any one operation bay of another yard crane. In Figure 8, the red 5, 4, and 8 are reversed. Then a real number between 0 and 1 is randomly generated, and when this number is smaller than the variation probability *r2*, the variation of the order of each yard crane operations is executed, i.e., two bays are randomly selected from the operational bays of each yard crane for position swapping, to obtain a new chromosome. In Figure 8, the green 2 and 8 switch places. 7 and 5,10 and 9 do the same.

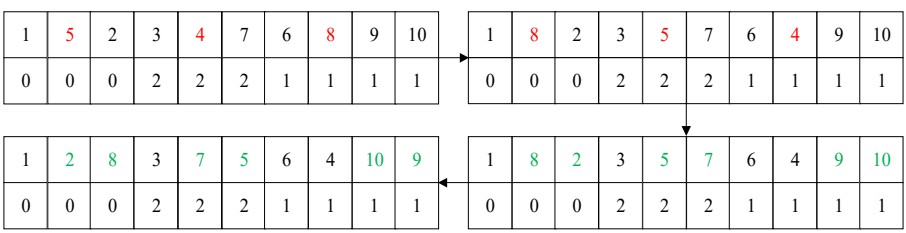

**Figure 8.** Chromosome variation diagram.

### 3.2.5. Genetic Repair Operation

Due to the randomness of the cross-variation process, there may be individuals among the chromosomes generated by the above operation whose yard cranes' cross-operation or the distance between yard cranes do not meet the safety distance, so these chromosomes that do not meet the constraints needed to be repaired.

The repair process is as follows: When the yard cranes' operations cross, the bays of the operating are swapped; when the distance between the yard cranes does not meet the safety distance, the bays to be operated that meet the conditions are retrieved and replaced to make them feasible solutions. If an individual cannot be repaired successfully, it needs to be deleted, and then the feasible solutions are randomly copied until the population size is reached.

## 4. Experiments and Discussion

### 4.1. Parameter Settings

Research on many ports shows that their pre-marshalling time is generally 3~8 h. The actual data of a port are used to analyze the calculation example in which a large ship is waiting for loading and unloading operation in a certain hub port, and the pre-marshalling period of the ship has also been determined. The focus is on a certain 30-bay export block with 300 containers to be loaded on the ship, all of which are stored in the odd stacks of the block. The even-numbered stacks in this area contain containers for other ships. Each container has a randomly given storage. There are eight stacks per bay in the block, with no more than four tiers.

The swarm size is now set to 20, the maximum number of attempts limit is 200, the maximum number of iterations is 1000, and the location of the containers in the 30 bays of this export block is randomly generated. The solutions are programmed using MATLAB R2014a on an Intel(R) Core(TM) i7-7700CPU@3.60GHz processor and a PC with 8 GB of RAM.

### 4.2. Experiments Analysis

After 20 solutions, the solution with the smallest value of the objective function and the best convergence result is selected, and the experimental results for 1–30 bays are obtained in turn, as shown in Table 2, to obtain 30 bays of yard crane operating time.

**Table 2.** Phase one result.

| NO. | CN | Obj (Min) | CPU Time (s) | NO. | CN | Obj (Min) | CPU Time (s) |
|-----|-----|-----------|--------------|-----|-----|-----------|--------------|
| 1 | 10 | 24 | 14.05 | 16 | 9 | 18 | 10.76 |
| 2 | 12 | 30 | 12.91 | 17 | 10 | 22 | 11.62 |
| 3 | 10 | 22 | 10.06 | 18 | 9 | 20 | 10.67 |
| 4 | 11 | 26 | 12.71 | 19 | 12 | 26 | 13.77 |
| 5 | 9 | 18 | 9.09 | 20 | 8 | 18 | 10.48 |
| 6 | 10 | 24 | 9.62 | 21 | 10 | 24 | 13.50 |
| 7 | 10 | 24 | 10.59 | 22 | 9 | 20 | 9.28 |
| 8 | 12 | 30 | 10.77 | 23 | 10 | 22 | 9.03 |
| 9 | 11 | 26 | 11.88 | 24 | 10 | 24 | 9.88 |
| 10 | 9 | 20 | 10.58 | 25 | 12 | 26 | 10.68 |
| 11 | 8 | 16 | 9.09 | 26 | 8 | 16 | 11.04 |
| 12 | 10 | 26 | 12.91 | 27 | 10 | 24 | 10.44 |
| 13 | 11 | 26 | 13.50 | 28 | 9 | 20 | 10.34 |
| 14 | 9 | 22 | 12.71 | 29 | 10 | 24 | 10.39 |
| 15 | 11 | 22 | 11.12 | 30 | 11 | 22 | 13.07 |

Note: CN—Number of target containers, Obj—Value of the objective function.

Using the data in Table 2 as known conditions, the solution of the second stage is carried out and the convergence results for the two yard cranes are shown in Figure 9a. The optimal solution is found after more than 400 iterations of the program and remained

constant, and the target converged at 345.67 min, taking a total of 14.82 s. The travel path of each yard crane is shown in Figure 9b, and the specific operational tasks and times for each yard crane are shown in Table 3.

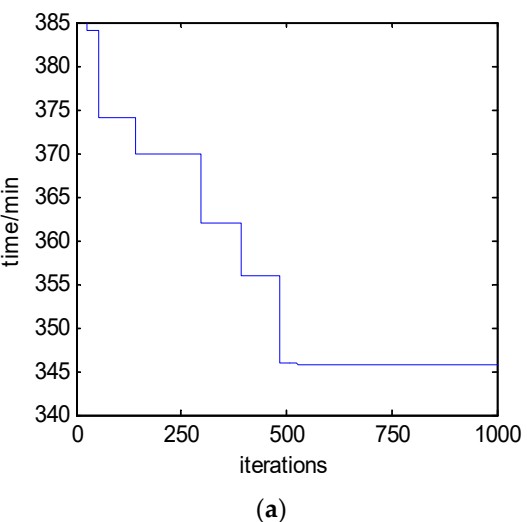

(**a**)

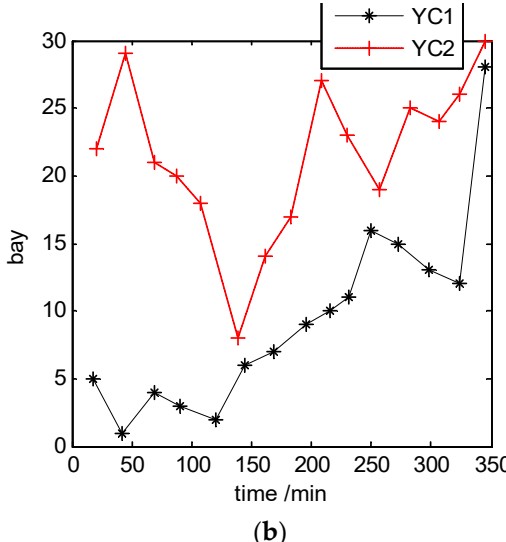

(**b**)

**Figure 9.** The solution of the second stage. (**a**) Convergence result of two yard cranes; (**b**) walking paths of two yard cranes.

**Table 3.** Operating hours of two yard cranes.

| NO. | Yard Crane1 | | | Yard Crane2 | | |
|---|---|---|---|---|---|---|
| | Bay | ST (Min) | FT (Min) | Bay | ST (Min) | FT (Min) |
| 1 | 5 | 0.00 | 18.00 | 22 | 0.00 | 20.00 |
| 2 | 1 | 18.33 | 42.33 | 29 | 20.58 | 44.58 |
| 3 | 4 | 42.58 | 68.58 | 21 | 45.25 | 69.25 |
| 4 | 3 | 68.67 | 90.67 | 20 | 69.33 | 87.33 |
| 5 | 2 | 90.75 | 120.75 | 18 | 87.50 | 107.50 |
| 6 | 6 | 121.08 | 145.08 | 8 | 108.33 | 138.33 |
| 7 | 7 | 145.17 | 169.17 | 14 | 138.83 | 160.83 |
| 8 | 9 | 169.33 | 195.33 | 17 | 161.08 | 183.08 |
| 9 | 10 | 195.42 | 215.42 | 27 | 183.92 | 207.92 |
| 10 | 11 | 215.50 | 231.50 | 23 | 208.25 | 230.25 |
| 11 | 16 | 231.92 | 249.92 | 19 | 230.58 | 256.58 |
| 12 | 15 | 250.00 | 272.00 | 25 | 257.08 | 283.08 |
| 13 | 13 | 272.17 | 298.17 | 24 | 283.17 | 307.17 |
| 14 | 12 | 298.25 | 324.25 | 26 | 307.33 | 323.33 |
| 15 | 28 | 325.58 | 345.58 | 30 | 323.67 | 345.67 |

Note: ST—Yard crane start time, FT—Yard crane completion time.

The number of yard cranes can be increased to improve operational efficiency. The travel path diagrams are shown in Figure 10a,b for the operation of three and four yard cranes, respectively. The travel path diagrams of the yard cranes clearly show that the yard cranes do not cross over during the operation and the solution is valid.

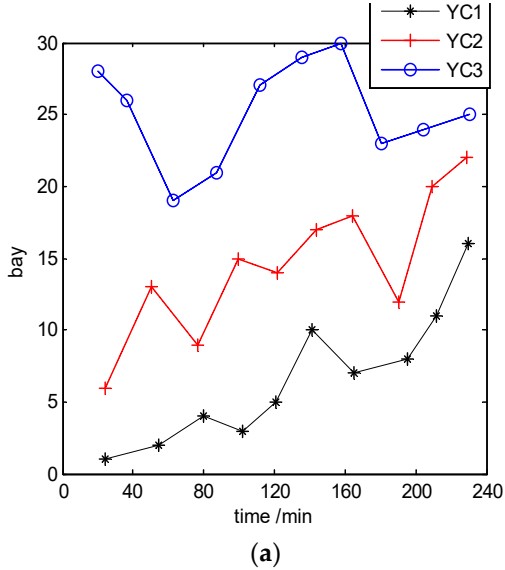 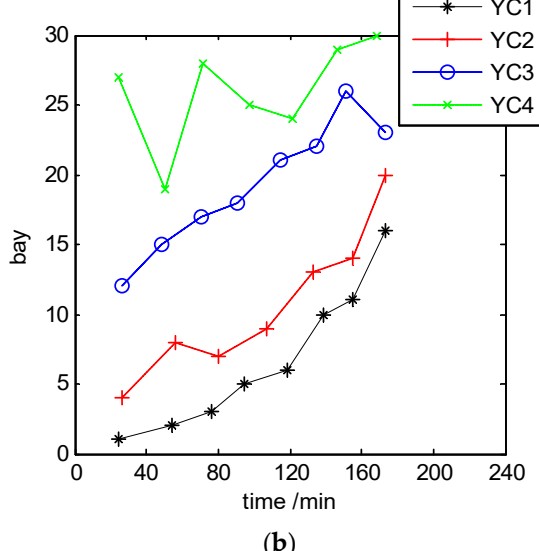

(**a**)              (**b**)

**Figure 10.** Multi-yard crane operational solutions. (**a**) Walking paths of three yard cranes; (**b**) walking paths of four yard cranes.

Table 4 shows that the completion time for two, three, and four yard cranes are 345.67 min, 230.42 min, and 173.50 min, respectively. When the pre-marshalling period is less than 6 h, the yard is best equipped with two yard cranes; when the pre-marshalling period is less than 4 h, three yard cranes are assigned optimally; when the pre-marshalling period is less than 3 h, four yard cranes are assigned most reasonably. The solution time shows that as the number of yard cranes increases, the more complex the problem becomes, and the longer the solution time becomes.

**Table 4.** Comparison of experimental results.

| YCN | Obj (Min) | YC1 (Min) | YC2 (Min) | YC3 (Min) | YC4 (Min) | CPU Time (s) |
|---|---|---|---|---|---|---|
| 2 | 345.67 | 345.58 | 345.67 | - | - | 14.82 |
| 3 | 230.42 | 229.92 | 229.17 | 230.42 | - | 18.25 |
| 4 | 173.50 | 173.25 | 173.50 | 173.42 | 168.25 | 19.33 |

Note: YCN, Number of yard cranes.

### 4.3. Validity Analysis

To verify the effectiveness of the algorithm, the results of the algorithm in this paper are compared and analyzed with the CPLEX solution. As there are non-linear constraints in both models, such as constraint 17 and constraint 38, they are relaxed so that the model is linearized and can be solved by CPLEX and used as a lower bound for the problem in this paper.

The data from the above examples are selected, and the results are compared by varying the number of operating bays and taking the average of the results of multiple calculations. As shown in Table 5, when the scale of the case is small, there is almost no deviation between the IABC-IGA and the lower bound obtained by CPLEX. As the scale of the case increases, the deviation gradually increases but does not exceed 5%, and when CPLEX is solved for 11 bays, it takes more than 90 min to find the optimal solution, and only a feasible solution can be given, thus verifying the effectiveness of the algorithm in this paper. In the actual operation, due to the limited pre-marshalling time available, the terminal usually needs to give a pre-marshalling operation solution in a short time, so the CPLEX solution is not feasible in real time.

**Table 5.** Solution quality of IABC-IGA and CPLEX.

| Bay Number | Containers Number | Obj | | GAP (%) | CPU Time | |
|---|---|---|---|---|---|---|
| | | IABC-IGA (Min) | CPLEX (Min) | | IABC-IGA (Min) | CPLEX (Min) |
| 7 | 2 | 88.33 | 88.33 | 0 | 7.59 | 0.016 |
| 8 | 2 | 100.42 | 100.42 | 0 | 7.75 | 0.088 |
| 9 | 2 | 114.67 | 112.58 | 1.82 | 8.22 | 1.16 |
| 10 | 2 | 126.15 | 122.31 | 3.04 | 8.63 | 19.45 |
| 11 | 2 | 139.74 | 132.95 * | 4.86 | 9.21 | 90 |
| 30 | 2 | 345.67 | - | - | 14.82 | - |

Note: * labels the feasible solution.

In order to verify the superiority and stability of the proposed algorithm, 20 tests are conducted on the data in Table 2. The worst solution, best solution, average value, and CPU time are given (see Table 6).

**Table 6.** Comparison of algorithms.

| Algorithm | IABC-IGA | GA | TS | GAP1 (%) | GAP2 (%) |
|---|---|---|---|---|---|
| Worst Solution (min) | 350.83 | 390.25 | 410.67 | 11.24 | 17.06 |
| Best Solution (min) | 345.67 | 355.50 | 354.83 | 2.84 | 2.65 |
| Average Value (min) | 347.25 | 376.86 | 388.73 | 8.53 | 11.95 |
| Average CPU Time (s) | 14.86 | 18.65 | 18.37 | 25.50 | 23.62 |

As can be seen from Table 6, all the solutions of IABC-IGA have better performance than that of GA and TS algorithms, with an average increase of 8.53% and 11.95%. Therefore, the algorithm in this paper is the best and most stable. In the 20 tests, the gap between the worst solution and the best solution of the TS algorithm is the largest, followed by GA, and the smallest by IABC-IGA. Thus, it can be seen that the first two algorithms are extremely unstable, and their solving quality is far lower than that of the algorithm in this paper. In terms of CPU Time, the solution speed of IABC-IGA is significantly higher than that of the other two algorithms.

*4.4. Scheme Comparison*

In order to further verify the effectiveness of the scheme, the scheme in this paper is compared with the current scheme. Through literature research, it is found that the minimum container relocation is used as the objective function of the model to calculate. The solution is conducted based on the example data in Table 2, and the comparison and analysis with the solution in this paper are shown in Table 7.

**Table 7.** Comparison of schemes.

| YCN | Yard Crane Completion Time | | GAP(%) |
|---|---|---|---|
| | This Article's Scheme (min) | Current Scheme (min) | |
| 1 | 666.92 | 728.42 | 9.22 |
| 2 | 345.67 | 375.17 | 8.53 |
| 3 | 230.42 | 246.75 | 7.09 |
| 4 | 173.50 | 183.94 | 6.02 |

It can be seen from Table 7 that the completion time of the yard crane in this paper is significantly less than that of the current scheme, and the optimization effect becomes more obvious when the number of yard cranes is smaller. Thus, the scheme in this paper is more in line with the actual situation and can effectively improve the operation efficiency of the

port, especially the small number of yard cranes and short idle time, and the advantages are more obvious.

### 5. Conclusions and Future Prospects

This paper studies the two-stage optimization problem of pre-marshalling operation and yard crane scheduling configuration in the export block and designs a two-stage mathematical programming model, which effectively improves the speed and quality of the solution. (1) For the problem of reshuffling in the bay, an improved artificial bee colony algorithm is designed, incorporating the principle of minimum overlapped containers for cutting the solution space and a genetic verification operation to ensure the solution's feasibility. (2) For the problem of yard crane allocation and scheduling within the block, an improved genetic algorithm is proposed, with a special strategy designed for improving the efficiency of the algorithm to generate the initial feasible solution and a gene repair operation designed for eliminating the infeasible solution. The design ideas of the algorithm in this paper can be used as a reference for solving similar problems. (3) Numerical experiments show that CPLEX can solve the exact solution when the number of bays is less than 11. When the number of bays reaches 30, CPLEX cannot solve the problem. The numerical results also verify that the solving quality of the IABC-IGA algorithm is improved by 8.53% and 11.95% on average compared with the conventional heuristic algorithm, with better performance.

In the future, it can be expanded from the following research directions. A more efficient and accurate algorithm can be designed to improve the solution efficiency. In addition, the operation optimization of multi-blocks under the influence of uncertain factors is considered to improve the overall operational efficiency of the yard.

**Author Contributions:** Conceptualization, S.D.; methodology, S.D.; software, S.D.; validation, H.Z.; resources, H.Z.; data curation, S.D.; writing—original draft preparation, S.D.; writing—review and editing, S.D., H.Z., and X.G. All authors have read and agreed to the published version of the manuscript.

**Funding:** This research was funded by the National Natural Science Foundation of China, grant number 71872025.

**Institutional Review Board Statement:** Not applicable.

**Informed Consent Statement:** Not applicable.

**Data Availability Statement:** The data presented in this study are available in the article.

**Conflicts of Interest:** The authors declare no conflict of interest.

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
