# Peer review of "Joint Optimization of Pre-Marshalling and Yard Cranes Deployment in the Export Block"

_processes, doi:10.3390/pr11020311_

Round 1

Reviewer 1 Report

Abstract:

- The abstract is very weak and needs significant improvement. 

- In line 19, you should include some one or two sentences  about the numerical results for the proposed methods.

Introduction:

From, line 93 to 98, the paper contributions are not clear, please highlight more these contributions clearly. Also, have you developed any of the proposed algorithms or jut you used the previous literature 

2.2. Mathematical Modeling

2.2.1. Model Assumption:

- This section is not clear, please add more clarifications and review the english language.

2.2.2. Variable Definitions  :

- Change to  " Mathematical notations" 

(1) Collection and parameters:

- Remove Collection.

2.2.3. Model Building:

- Change it to " Mathematical formulation" 

- Also, ij the objective function, could you please extend or identify the terms of the  Tl : Total operating time of the yard crane at l th bay. 

 IABC Algorithm Pseudocode

- The font size needs to be consistent in the manuscript.

- Pseudocode needs improving 

5. Conclusions and Future Prospects

Please extend the conclusions with highlighting on some computational results. 

References

- This section needs to be updated inclsuing 2021, 2022.

Reviewer 2 Report

It would be really helpful for the readers if the authors can present a small case study showing how the proposed algorithm improves the proposed operational efficiency of the port. Overall, the work done is good.

Reviewer 3 Report

Paper is really well written regarding all criteria and my only request is to explain why  the general algorithm does not solve large-scale cases (line 75). It could be published after that.

Author Response

Thanks very much for the comments of the reviewer.

The author did not express clearly, and it had been changed to general accurate algorithms cannot solve large-scale experiments.

Reviewer 4 Report

This paper proposed a two-stage mathematical model for pre-marshaling and yard crane scheduling problems to minimize the operation time of the yard crane relocation and minimize the maximum completion time. Two metaheuristic techniques were used to solve the two-stage mathematical model: an artificial bee colony algorithm for the first-stage model and an improved genetic algorithm for the second-stage model. The authors verify the effectiveness of the proposed algorithm by comparing the proposed algorithm with the exact algorithm and other classical metaheuristic methods. The problem and proposed algorithm are interesting, but this article needs further improvement for more clarity and ease for the reader to understand. There are some revisions as follows:

- Why not integrate both problems (pre-marshaling and yard crane scheduling problems) and solve both problems in one stage? The authors should explain the reasons for presenting the two-stage mathematical model.

- Section 2.1 need to be rewritten to be clearer and easier to understand. These two articles [1] - [2] may help to provide guidelines for describing the problem.

1.          Tanaka, S.; Tierney, K. Solving real-world sized container pre-marshaling problems with an iterative deepening branch-and-bound algorithm. European Journal of Operational Research 2018, 264, 165–180, doi:10.1016/j.ejor.2017.05.046.

2.          Galle, V.; Barnhart, C.; Jaillet, P. Yard Crane Scheduling for container storage, retrieval, and relocation. European Journal of Operational Research 2018, 271, 288–316, doi:10.1016/j.ejor.2018.05.007.

- Figure 1 is unclear and may confuse the reader. Please verify.

- In section 2.2.2, suggest separating indices, parameters, and decision variables.

- Suggest explaining the link between the first and second stages of the problem in section 2.2.3.

- Authors should add equations to describe Tl and Tk.

- Are indices i and j redundantly defined or not? Please verify.

- A description of the meaning of YC should be added to the notes in Table 3.

- Table 6: In my opinion, the effectiveness of the proposed algorithms differs slightly from GA and TS. The authors should discuss whether the slight difference is significant enough to explain how the IABC-IGA method is more effective than GA and TS.

Round 2

Reviewer 1 Report

The authors answered the proposed comments. 

Reviewer 4 Report

After modifications, I think the manuscript can be accepted.